# TornadoAttention: Hardware-Efficient Sparse Attention via Fine-Grained Spatio-Temporal Permutation

## Abstract

Diffusion Transformers (DiTs) have demonstrated remarkable success in video generation. However, their core component, the self-attention mechanism, suffers from quadratic complexity. To alleviate this issue, sparse attention mechanisms have been proposed. Existing methods, however, often impose strong, handcrafted priors, restricting attention to a few fixed patterns, fail to capture the diverse, data-dependent attention patterns unique to each layer and head. Motivated by the spatio-temporal locality of self-attention in DiTs, we propose TornadoAttention, a **training-free** sparse attention mechanism. Our key idea is to apply a fine-grained permutation of the query and key sequences that better matches the underlying attention structure. We applied TornadoAttention to advanced open-source video generation models. It reveal that the attention masks obtained through offline searching exhibit excellent generalization capabilities across a diverse range of prompts, which provides a crucial foundation for aggressive, hardware-specific kernel-level optimizations. On the HunyuanVedio model, our method achieves a $1.4\times$ speedup with negligible loss in fidelity.

## 1 Introduction

Amidst the rapid advancement of Artificial Intelligence Generated Content (AIGC) technologies, video generation, e.g., Text-to-Video (T2V) (Hong et al., 2022; Villegas et al., 2022; Wang et al., 2023a) and Image-to-Video (I2V) (Wang et al., 2023b; Zhang et al., 2023a), has emerged as a prominent research frontier. These technologies are designed to automatically synthesize realistic video sequences from user-provided textual descriptions or static images. The Diffusion Transformer (DiT) (Ma et al., 2024; Peebles & Xie, 2023) architecture has demonstrated exceptional performance. Self-attention mechanism within the DiT models the interactions among these tokens.

However, for high-resolution or long-duration videos, the length of the token sequence ($n$) can become exceedingly large. For instance, in the Wan2.1-14B model (Wan et al., 2025), a three-second video at 720p resolution comprises 75,600 tokens. The standard self-attention mechanism in DiT, which computes the relationships between every token and all others in the sequence, exhibits computational and memory complexities proportional to the square of the sequence length ($O(n^2)$). In typical DiT-based video generation models, the self-attention computation accounts for over 70% of the end-to-end latency (Chen et al., 2025). Consequently, the attention computation within the DiT framework has emerged as the primary bottleneck impeding efficient model inference.

To address the bottleneck imposed by the standard self-attention mechanism, researchers have proposed sparse attention mechanisms (Tan et al., 2025), which restrict each token's attention to a carefully chosen subset of informative tokens rather than all tokens. In video DiT models, attention maps naturally show high sparsity: for instance, some heads focus on local spatial details, while others capture temporal correlations (Xi et al., 2025). These patterns motivate the design of more efficient attention mechanisms. Current work on sparse attention for video DiTs generally falls into two main approaches:

- Fixed Patterns: This approach utilizes a predefined set of static sparse connection patterns, which are formulated based on prior knowledge or offline statistical analysis of attention

(a) Unclassifiable Patterns  (b) Unclassifiable Patterns  (c) Intra-Pattern Variation  (d) Intra-Pattern Variation  (e) Intra-Pattern Variation  (f) Query-Dependent Sparsity

Figure 1: Examples of Diverse Attention Patterns. These include (a, b) complex patterns that defy simple classification, (c, d, e) significant variations within the same pattern family (e.g., multi-diagonal), and (f) query-dependent sparsity.

map. For example, observing that attention often concentrates on spatially local regions or specific temporal steps, researchers have designed sparse masks such as main diagonals (for attending to neighboring regions), multi-diagonals (for capturing local information at different scales), or vertical/horizontal stripes (where specific tokens serve as global information hubs). This approach is simple and hardware-friendly but its rigidity and strong priors can degrade video quality in certain case.

- Dynamic Patterns: In contrast to fixed patterns, this approach operates on the premise that important attention regions are dynamic and content-aware. It selects sparse regions via a "coarse-to-fine" process: (1) downsample query and key in self-attention to build a representative attention map; (2) locate high-score regions; (3) compute full attention only within these regions (Shen et al., 2025; Yang et al., 2025). This method is more flexible and adapts to varying video content, but adds online overhead to search for sparse patterns.

Different attention heads often follow patterns such as main-diagonal, multi-diagonal, triangular, or vertical-line configurations. Yet relying solely on these predefined masks is insufficient, as they frequently miss high-value attention regions in many cases, as illustrated in Figure 1.

- Complex, Unclassifiable Patterns: Some attention maps do not conform to any simple, classifiable geometric pattern.

- Intra-Pattern Variation: The specific shape within a given pattern type can vary. For instance, in a multi-diagonal pattern, the width of the diagonals can differ among heads or even within the same head (Li et al., 2025b).

- Query-Dependent Sparsity: We observe that within certain heads, different query tokens exhibit varying degrees of concentration in their attention distributions over key tokens. Some query tokens have highly concentrated attention on a few key tokens, while others have more dispersed attention.

Motivated by the diversity of attention patterns and the architectural preferences of GPUs, we propose TornadoAttention, a training-free and prompt-independent sparse attention scheme designed to accelerate DiT-based video generation models. Our approach is twofold. First, to capture the spatio-temporal locality, we partition the 3D sequence along each of its temporal ($T$), height ($H$), and width ($W$) dimensions, thereby forming local 3D tiles. Second, we devise a four-dimensional search space to identify the optimal token permutation scheme. This search space is defined by: 1) the permutation order of the dimensions, and 2, 3, 4) the block sizes for the height, width, and time dimensions. Subsequently, for each candidate configuration, we construct an attention mask by starting from the main diagonal of the full attention map and symmetrically expanding outwards until a predefined threshold for the percentage of retained attention scores is reached. Finally, among all evaluated configurations, we select the permutation scheme and its corresponding attention mask with the highest sparsity.

The main contributions of our work are summarized as follows:

- We propose TornadoAttention, a novel sparse attention mechanism that, instead of merely selecting important tokens, proactively reorders the token sequence via a fine-grained spatio-temporal permutation to align the memory layout with the natural locality of video data.

- To the best of our knowledge, we are the first to propose a **training-free** sparse attention scheme for video DiTs that achieves prompt-agnostic acceleration solely through an **offline search** process. This enables an out-of-the-box deployment model where a one-time search yields a universally effective static mask.

- We demonstrate the effectiveness of TornadoAttention on state-of-the-art video generation models, achieving a significant 1.4x speedup on the HunyuanVedio-T2V-14B model with negligible degradation in generation quality, validating our approach as a practical solution for accelerating large-scale video DiTs.

## 2 PRELIMINARY

**Self-Attention in Video DiTs**. In classic video generation of DiT, a video clip into a compressed latent representation $z \in \mathbb{R}^{T \times H \times W}$ using a pre-trained Variational Autoencoder (VAE). This 3D tensor is then flattened along its spatio-temporal dimensions into a sequence of $N = T \times H \times W$ tokens, forming the input $X \in \mathbb{R}^{N \times D}$ for a Transformer head. The self-attention mechanism models the dependencies between these tokens. From the input sequence $X$, learnable linear projections generate the Query ($Q$), Key ($K$), and Value ($V$) matrices. An $N \times N$ attention matrix $A$, also called attention map, which quantifies the pairwise relevance between all tokens, is then computed via scaled dot-product and a softmax, $D$ is the hidden dimension of the queries and keys.

$$A = \text{softmax}\left(\frac{QK^T}{\sqrt{D}}\right) \tag{1}$$

The final output is a weighted sum of the Value vectors, computed as $O = AV$. The primary computational bottleneck is the calculation and storage of this dense matrix $A$, which scales quadratically with the sequence length, $\mathcal{O}(N^2)$.

**Sparsification Attention**. The attention matrix, or also called attention map, $A$, often demonstrates sparsity in video DiTs: only a small fraction of its entries contain high attention scores. The objective of sparse attention is to approximate the full attention output by computing only a small, carefully selected subset of the most significant scores. Formally, let $S$ be the set of all possible token index pairs $(i, j)$ where $1 \leq i, j \leq N$. Our goal is to identify a subset of indices $S_{sparse} \subset S$,

Different methods vary in how they determine the subset. Common strategies include selecting a fixed number of entries per query or adhering to a predefined sparsity (e.g., 20% of the total entries). However, these methods provide limited control over information loss, as the distribution of attention scores can vary dramatically across heads and inputs.

We adopt a more principled, energy-based criterion analogous to top-$p$ (nucleus) sampling. Our goal is to find the smallest possible set of token pairs, $S_{sparse}$, whose cumulative attention score meets a target energy threshold $\tau$ (e.g., $\tau = 0.9$). This ensures that a consistent percentage of the model's learned context is preserved. Formally, we define the optimal sparse set $S_{sparse}$ as the solution to the following constrained optimization problem:

$$S_{sparse} = \underset{S' \subseteq S}{\text{argmin}} \, |S'| \quad \text{subject to} \quad \sum_{(i,j) \in S'} A_{ij} \geq \tau \sum_{(i,j) \in S} A_{ij} \tag{2}$$

where $S$ is the set of all possible token index pairs. The challenge, therefore, is not just to satisfy this condition, but to find a way to structure the resulting set $S_{sparse}$ into a hardware-efficient pattern. Our method, detailed in Section 4, focuses on finding a permutation strategy that makes this optimally compact, energy-based set $S_{sparse}$ computationally feasible.

## 3 RELATED WORK

Using the sparsity of attention to speed up computation has been proved to be accessible by recent researches. In LLMs, LM-Infinite (Han et al., 2023) and StreamingLLM (Xiao et al., 2023) find temporal locality. H2O (Zhang et al., 2023b) and InfLLM (Xiao et al., 2024a) adopt sliding window

pattern while SampleAttention (Zhu et al., 2024) and MOA (Fu et al., 2024) also use attention sink pattern. DuoAttention (Xiao et al., 2024b) and MInference (Jiang et al., 2024) preset different patterns for different heads. At present, two kinds of methods are mainly used to accelerate computing by using attention sparsity. The training-free method does not need to modify the existing model, but only accelerates the video generation. The training method is added when the model is trained, which can reduce redundant calculation from training.

**Training-Free Approaches.** A common strategy to accelerate attention in video generation is to match attention heads with a small set of predefined sparse patterns and construct masks accordingly. STA (Zhang et al., 2025b) introduces sliding tile attention to speed up pattern matching. Sparse VideoGen (SVG1) (Xi et al., 2025) introduces temporal and spatial sparse attention patterns and performs binary classification via sampling. Radial Attention (Li et al., 2025b) further observes that both patterns decay exponentially with their spatial or temporal token distance, and unifies them into a single dynamic pattern. Sparse-vDiT (Chen et al., 2025) identifies an additional global vertical-line pattern and VORTA (Sun et al., 2025) categorizes heads into detail-, semantic-, and global-level groups and pre-trains a router for head assignment.

Other works argue that 2D attention maps disrupt spatiotemporal coherence and reformulate them to preserve temporal consistency. Most of these approaches summarize limited combinations for a particular model and often rely on offline computation. PAROAttention (Zhao et al., 2025) reshapes the $t, w, h$ dimensions into $h, w, t$, concentrating the attention distribution, while CompactAttention (Li et al., 2025a) restores the 3D structure and progressively shrinks masks offline to obtain suitable patterns. These methods highlight recurring structures in attention maps and demonstrate the potential of 3D compression, yet they do not fully exploit both advantages simultaneously. Our method addresses this by unifying spatiotemporal correlations in 3D space, leveraging reshaping to enhance efficiency while fully considering the diversity of attention in spatial and temporal distribution.

Other training-free approaches avoid preset patterns and instead reduce semantic redundancy through online computation. AdaSpa (Xia et al., 2025) locates self-similar attention blocks for dynamic masking and accelerates retrieval with an LSE-based cache. DraftAttention (Shen et al., 2025) estimates high-attention-score regions via a low-resolution attention map. Sparge Attention (Zhang et al., 2025a) compresses highly redundant query and key blocks into single tokens. Jenga (Zhang et al., 2025d) partitions tokens by 3D spatial distance and restricts computation to strongly correlated key-value pairs. These approaches offer more flexible masks but incur higher online overhead compared to fixed-pattern methods.

**Training-Based Approaches.** Another line of research incorporates sparsity directly during training to reduce overall cost. The central idea is to compute only a small subset of key blocks that dominate the final results. DSV (Tan et al., 2025) coarsely estimates key-value attention scores and computes only the highest ones. VMoBA (Wu et al., 2025) partitions key blocks by different layers and dynamically selects similar key-value pairs. Bidirectional Sparse Attention (BSA) (Zhan et al., 2025) extends this by also optimizing key-value blocks, and discard semantic similar tokens in query blocks. Native Sparse Attention (NSA) (Yuan et al., 2025) jointly optimizes keys and values via compression, token selection, and sliding-window strategies. Unlike training-free methods, these approaches modify model parameters and require costly fine-tuning, limiting generalization across different structures.

**Emerging Trends**. An important trajectory can be observed in sparse attention for video DiTs: the field is moving from rigid, prior-driven fixed patterns toward dynamic and fine-grained adaptive strategies, which is consistent with our improvement. This shift is well illustrated by the evolution from SVG 1 to SVG 2 (Yang et al., 2025). In SVG1, once a head was classified as "spatial", all its queries were constrained to within-frame Keys. SVG2 refined the granularity from head-level to token-cluster-level, clustering tens of thousands of tokens into a few hundred semantic groups, and performing fine-grained attention only among the most critical clusters. This reflects a broader trend toward adaptivity and finer search resolution.Building on this trajectory, our work further advances sparse attention by integrating spatiotemporal coherence with structural flexibility, enabling both efficient computation and stronger generalization across diverse video generation scenarios.

Figure 2: The Misalignment of Logical Locality and Memory Layout caused by Standard Raster-Scan Flattening. A 3D spatio-temporal volume of video frames (left) is converted into a 1D token sequence (right) As highlighted by the red boxes, tokens that are adjacent in the original 3D space, both spatially (e.g., different parts of the goose's body within the same frame) and temporally (e.g., the goose's head across consecutive frames), become widely separated in the 1D sequence.

## 4 METHODOLOGY

Our proposed method, TornadoAttention, addresses the computational bottleneck of the self-attention mechanism in video DiTs. The core idea is not to select high attention scores, but to reorder the input token sequence in a way that aligns with the natural spatio-temporal locality of video data.

### 4.1 THE PRINCIPLE OF SPATIO-TEMPORAL LOCALITY

Video data is inherently structured and exhibits strong spatio-temporal locality. This principle arises from two fundamental properties of the physical world:

**Spatial Locality**: Objects are spatially coherent. A pixel or image patch is most strongly correlated with its immediate spatial neighbors, as they collectively form textures, edges, and surfaces.

**Temporal Locality**: The world evolves continuously over time. Consecutive video frames typically depict the same scene with minor changes. Consequently, a patch at time $t$ is highly correlated with the patch at or near the same spatial location at time $t + 1$.

In a standard DiT, where the $(T, H, W)$ latent space is flattened into a 1D sequence using a fixed raster-scan order as show in Figure 2. The self-attention mechanism learns to approximate the fundamental spatio-temporal correlations, and the resulting attention patterns directly reflect the scope of the locality captured by a given attention head. A strong **main-diagonal** pattern, for instance, reflects immediate spatial locality. The characteristic **multi-diagonal** patterns capture temporal locality by attending to corresponding patches in adjacent frames. A **lower-triangular** (causal) pattern signifies attention over the entire spatio-temporal past, while a **banded** (sliding-window) pattern focuses on a more constrained local neighborhood in recent time and space. Finally, a **fully dense** pattern represents attention over the entire global context. Crucially, while all these patterns are logical manifestations of locality, the fixed flattening order forces them into computationally inefficient forms.

### 4.2 TORNADOATTENTION: PERMUTATION SEARCH FOR LOCALITY ALIGNMENT

Instead of passively accepting the default token order, we proactively search for an optimal, head-specific permutation $P$ that rearranges the token sequence. The goal is to ensure that tokens with high mutual attention scores are placed adjacently in the new permuted sequence. This process effectively gathers the scattered attention and concentrates them around the main diagonal.

#### 4.2.1 THE PERMUTATION SEARCH SPACE

Our search for the optimal permutation is conducted over a highly structured and meaningful space of possibilities. As shown in Figure 3, our search space defined by two key components:

1. Tile Granularity ($t_{size}$, $h_{size}$, $w_{size}$): This defines the size of the elementary spatio-temporal cubes that we reorder. By varying these dimensions (e.g., $t_{size} = \{1, 1/2, 1/4, 1/8\}$), we can capture locality at different scales, allowing some permutations to prioritize fine-grained spatial details while others focus on longer temporal dynamics.

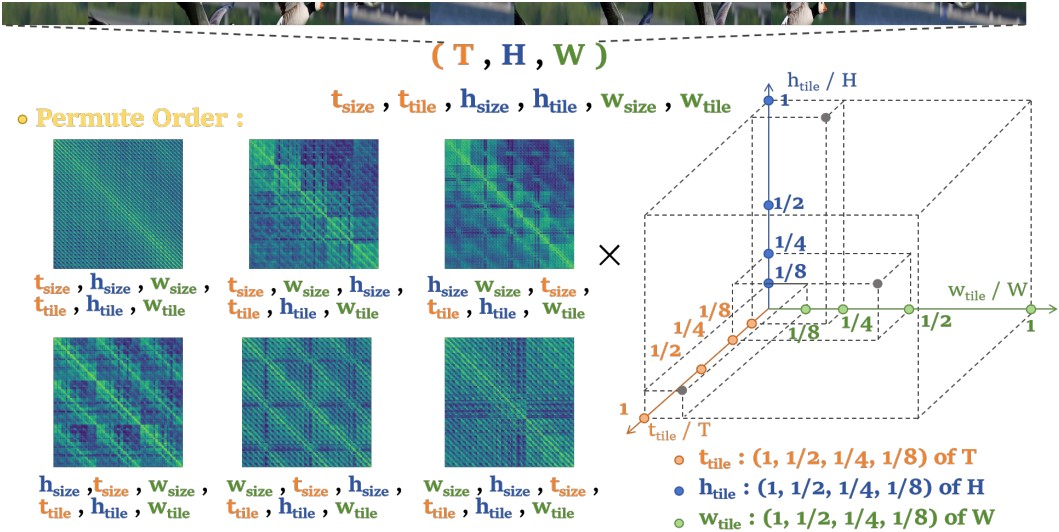

Figure 3: The Structured Search Space for the TornadoAttention. (Left) The Permutation Order, which defines the hierarchical arrangement of tokens. We explore various einops-style permutations of the six factors corresponding to number of tiles ($t_{size}, h_{size}, w_{size}$) and their respective tile sizes ($t_{tile}, h_{tile}, w_{tile}$). (Right) The Block Granularity, which determines the size of the elementary blocks being rearranged. The final search space is the Cartesian product of these candidate orders and granularities.

2. Permutation Order (*new shape*): This defines the hierarchical order in which the elementary blocks and the larger "tiles" they form are arranged in the final 1D sequence. We use *einops*-style string notation (e.g., "$h_{size}, w_{size}, t_{size}, t_{tile}, h_{tile}, w_{tile}$") to represent these orders. This notation specifies the nesting of loops for rearranging the data; for instance, placing $t_{tile}$ in the innermost position prioritizes temporal continuity, clustering tokens from consecutive frames together.

The full search space $\Pi$ is the Cartesian product of all candidate block granularities and permutation orders, providing a rich yet manageable set of potential spatio-temporal data layouts. We have implemented an efficient search algorithm that employs several tricks, including pruning duplicate permutations on the fly. The technical details are provided in Appendix A.1.

### 4.2.2 OPTIMIZATION OBJECTIVE AND SEARCH PROCESS

Since the same permutation $P$ is applied to both queries and keys, tokens that are close in the original 3D space and thus likely to have high mutual attention are brought to similar indices in the 1D sequence, causing their high attention scores to manifest along the main diagonal. For each attention head, our objective is to find the permutation $P^* \in \Pi$ that makes the resulting attention matrix $A'$ maximally concentrated around its main diagonal. We quantify this concentration with a metric called the Concentration Ratio ($\mathcal{R}_C$), where a lower value indicates better concentration.

The search process to find the optimal permutation $P^*$ for a given attention head is performed as follows, using a pre-computed or sampled attention matrix $A$ as a proxy:

1. Iterate through Permutations: For each candidate permutation recipe $\pi \in \Pi$, we define a corresponding permutation matrix $P_\pi$. This matrix is applied to the rows and columns of the attention matrix $A$ to obtain the permuted matrix $A' = P_\pi A P_\pi^T$.

2. Define Computational Block Structure: Inspired by STA (Zhang et al., 2025c), to align with hardware constraints (e.g., memory coalescing), the permuted matrix $A'$ is conceptually divided into a $B \times B$ grid of computation blocks. We fix $b = N/B = 128$.

3. Determine the Block-Row Masks: We construct the optimal sparse mask $\mathcal{M}_\pi^*$ by performing an independent search for each of the $B$ block rows. For a given block row $i \in [1, B]$: a. Calculate

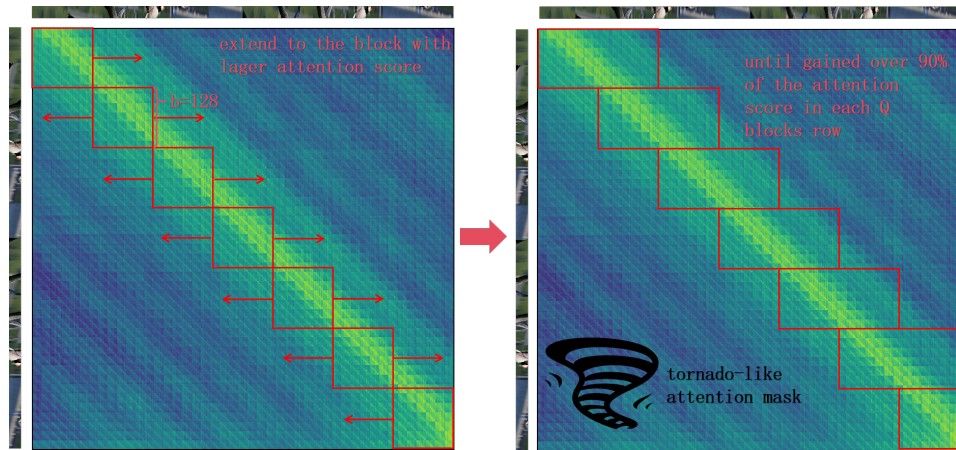

Figure 4: Search for the Attention Mask in One Head. (Left) The search begins with a mask containing only the main diagonal blocks. The process then iteratively extends the mask on a per-row basis, always selecting the adjacent block (left or right) with the highest attention score. (Right) This expansion continues until the cumulative score within the masked region for each block-row surpasses the 90% threshold.

the total energy $E_{row}(i)$ contained within all blocks $(i,j)$ for $j \in [1,B]$. b. Start with an initial mask for this row, $\mathcal{M}_0^{(i)} = \{(i,i)\}$, containing only the diagonal block. c. Iteratively expand the row mask $\mathcal{M}^{(i)}$ by adding the adjacent block (either $(i, j-1)$ or $(i, j+1)$) with the highest energy. d. Terminate the expansion for row $i$ when the cumulative energy within its mask $\mathcal{M}^{(i)}$ reaches the threshold $\tau$ (e.g., 0.9) of $E_{row}(i)$. The final global mask for the permutation $\pi$ is the union of all row masks:

$$\mathcal{M}_\pi^* = \bigcup_{i=1}^{B} \mathcal{M}^{(i)} \tag{3}$$

4. Calculate Concentration Ratio: The concentration ratio $\mathcal{R}_C(\pi)$ is defined as the average number of blocks selected per row. Mathematically, this is the total number of blocks in the final mask divided by the number of rows $B$:

$$\mathcal{R}_C(\pi) = \frac{|\mathcal{M}_\pi^*|}{B} \tag{4}$$

This metric directly reflects the average "bandwidth" (in blocks) required to capture $\tau$ of the attention energy across all query blocks.

5. Select Optimal Permutation: We select the permutation $\pi^*$ that minimizes this concentration ratio. This ensures we find the layout that requires the narrowest, most hardware-efficient computation band:

$$\pi^* = \operatorname*{argmin}_{\pi \in \Pi} \mathcal{R}_C(\pi) = \operatorname*{argmin}_{\pi \in \Pi} \frac{|\mathcal{M}_\pi^*|}{B} \tag{5}$$

This offline profiling stage yields, for each Transformer head, an optimal permutation $P_{\pi^*}$ and a corresponding fixed block-sparse mask $\mathcal{M}_{\pi^*}^*$, which are then used for inference.

# 5 EXPERIMENTS

## 5.1 EXPERIMENTS SETUP

### 5.1.1 IMPLEMENTATION DETAILS

All experiments were conducted on 8 NVIDIA H100 GPUs with CUDA 12.8. To ensure a fair and direct comparison with current state-of-the-art methods, we integrated our proposed TornadoAtten-

tion mechanism directly into the open-source SVG codebase. The underlying block-sparse attention computation is efficiently handled by leveraging the flexible attention interfaces provided by the Magi Attention library (Zewei & Yunpeng, 2025). The key hyperparameters for TornadoAttention, including the candidate sets for block granularities and used in our search, are detailed in Table 1.

Table 1: Configurations of TornadoAttention w.r.t. Hunyuan-T2V-14B and Wan2.1-T2V/I2V-14B.

| Hyper-parameter | Hunyuan-T2V-14B | Wan2.1-T2V/I2V-14B |
|---|---|---|
| num_frame | 129 | 80 |
| frame_size | $1280 \times 720$ | $1280 \times 720$ |
| latent shape of video | (33, 45, 80) | (21, 45, 80) |
| Candidates of $t_{\text{tile}}$ | {1, 3, 11, 33} | {1, 3, 7, 21} |
| Candidates of $h_{\text{tile}}$ | \multicolumn{2}{c}{{1, 4, 8, 16, 45}} |
| Candidates of $w_{\text{tile}}$ | \multicolumn{2}{c}{{1, 4, 8, 16, 80}} |
| $\tau$ (top-p) | \multicolumn{2}{c}{0.9} |

For the HunyuanVideo model, following the settings of SVG, we retain dense attention for the text-embedding part. Furthermore, consistent with previous methods (Shen et al., 2025), we apply dense attention for the initial 25% of the denoise timesteps for all models.

### 5.1.2 MODELS AND DATASETS

We evaluate the effectiveness of our method on a suite of large-scale, state-of-the-art video generation models to demonstrate its broad applicability. The models include HunyuanVideo-T2V-14B (Kong et al., 2024), Wan2.1-T2V-14B (Wan et al., 2025) for text-to-video (T2V) tasks, and Wan2.1-I2V-14B for image-to-video (I2V) tasks. For T2V generation experiments, we utilize prompts from Penguin Benchmark in HunyuanVideo, a standard dataset designed to evaluate the creative and compositional capabilities of video models.

### 5.1.3 EVALUATION METRICS

Our evaluation is twofold, assessing both the fidelity of our approximation and the perceptual quality of the final video output. Fidelity vs. Dense Attention: To quantify how well our sparse attention mechanism preserves the output of the original model, we measure the frame-by-frame difference between videos generated with TornadoAttention and those from the dense-attention baseline. We report three standard metrics: Peak Signal-to-Noise Ratio (PSNR), Structural Similarity Index Measure (SSIM), and Learned Perceptual Image Patch Similarity (LPIPS) (Zhang et al., 2018).

Generation Quality: Following established best practices in video generation evaluation, we also assess the perceptual quality of the generated videos themselves. We employ the comprehensive VBench benchmark (Huang et al., 2024), which evaluates various aspects such as temporal consistency, motion quality, and object fidelity.

### 5.1.4 BASELINES

Our primary baseline for comparison is SVG, a recent and highly competitive open-source method for efficient video generation. By implementing TornadoAttention within the SVG framework, we enable a direct and controlled comparison of the attention mechanisms.

### 5.2 MAIN RESULT

Table 2 presents the primary results of TornadoAttention compared to the Sparse VideoGen (SVG) baseline on both Hunyuan-T2V and Wan2.1-T2V/I2V models. Our method achieves a significant 1.4x speedup on the Hunyuan-T2V (720p) task, as indicated by the reduction in PFLOPS. Crucially, this acceleration is achieved with negligible degradation in video quality. This is evidenced by the PSNR, SSIM and LPIPS scores, which remain nearly identical to the dense attention baseline

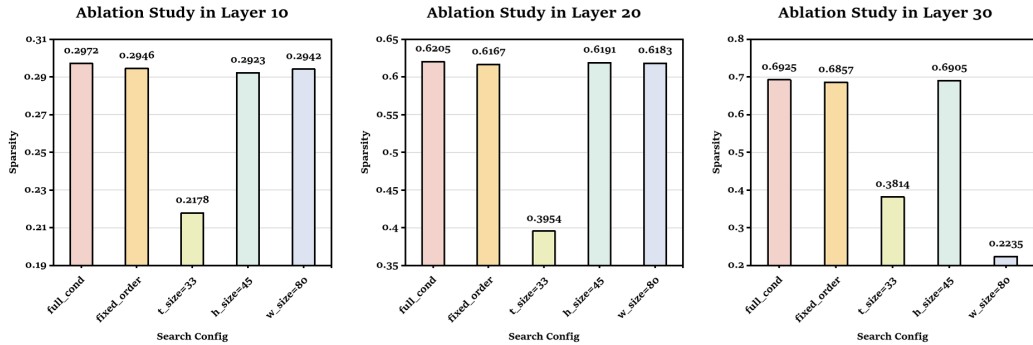

Figure 5: Ablation study on sparsity at layers 10, 20, and 30 under different search configurations. Our strategy with all four dimensions(full_cond) maintains the highest sparsity, while other searches degrade it.

implemented in SVG. The results on VBench metrics, further confirm that our sparse approximation does not compromise the model's generative capabilities. These findings validate our hypothesis that reordering tokens to match the inherent spatio-temporal locality is a highly effective strategy for creating hardware-efficient, static sparse patterns.

Table 2: Main results of the proposed method compared to the Sparse VideoGen (SVG). The end-to-end latency of video generation is averaged on 10 videos.

| Model | Method | PSNR ↑ | SSIM ↑ | LPIPS ↓ | Img. Qual. | Sub. Cons. | Bakg. Cons. | Dyn. Deg. | Aes. Qual. | Latency(s) ↓ |
|---|---|---|---|---|---|---|---|---|---|---|
| Hunyuan-T2V (720p) | SVG | 31.12 | 0.9337 | 0.0600 | 0.6611 | 0.9179 | 0.9547 | 0.4000 | 0.5535 | 1342 |
| | Ours | 25.18 | 0.8361 | 0.1863 | 0.6569 | 0.9285 | 0.9583 | 0.4000 | 0.5509 | 1151 |

## 5.3 ABLATION STUDY

To validate the effectiveness of our multi-dimensional search space, we conducted an ablation study, with the results for layers 10, 20, and 30 presented in Figure 5. The study compares the sparsity achieved by our full four-dimensional search (full_cond) against several constrained search configurations: fixing the permutation order, and searching without dimensions of $T$, $H$ and $W$.

As the figure clearly illustrates, the full_cond strategy consistently discovers the sparsest attention masks (i.e., achieves the highest sparsity values) across all tested layers. This demonstrates that jointly optimizing the permutation order and the block granularities across all three (T, H, W) axes is crucial. Restricting the search space significantly limits the ability to find an optimal data layout that concentrates attention scores, leading to denser and less efficient masks. This validates our design choice of a comprehensive offline search space to maximize computational savings at inference time.

## 6 CONCLUSION

In this work, we addressed the critical computational bottleneck in Diffusion Transformer (DiT) models for video generation. We introduced TornadoAttention, a novel, training-free, and prompt-independent sparse attention scheme. Our core contribution is the principle of proactively reordering tokens via a fine-grained spatio-temporal permutation to align the data layout with the inherent locality of video data. Our method achieves a 1.4x speedup on the state-of-the-art HunyuanVedio model with negligible loss in fidelity. Future work will explore hybrid approaches to capture semantic, non-local patterns and pursue further kernel-level optimizations to fully leverage the static nature of our generated masks.

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

# A  APPENDIX

## A.1  DETAILS OF THE EFFICIENT PERMUTATION SEARCH ALGORITHM

The search space for the optimal permutation, while structured, is vast. On a large model such as Hunyuan-T2V-14B, a naive, brute-force search through all candidate permutations can take approximately 30 minutes per attention layer. To make this offline profiling stage practical and scalable, we designed an efficient search algorithm incorporating three key optimization techniques: (1) search space pruning to eliminate redundant computations, (2) an early exit strategy during mask evaluation, and (3) a heuristic search order to maximize the effectiveness of the early exit.

**1. Search Space Pruning.**  The Cartesian product construction of our search space $\Pi$ introduces significant redundancy. This occurs because different combinations of block granularities and permutation orders can result in functionally identical token permutations. For example, if we choose a temporal block size equal to the full temporal length ($t_{size} = T$), the corresponding $t_{tile}$ factor has a dimension of one. In this case, any two new shape strings that only differ by the relative order of $t_{size}$ and $t_{tile}$ (e.g., $...t_{size}...t_{tile}...$ vs. $...t_{tile}...t_{size}...$) will produce the exact same permutation.

To avoid re-evaluating these duplicates, we implement a pruning mechanism based on a canonical representation of each permutation. For each permutation recipe $\phi \in \Pi$, we first compute its unique signature by applying it to a standard index tensor $I = [0, 1, \ldots, N-1]$. The resulting permuted tensor, $I' = \phi(I)$, serves as a unique identifier for the permutation's effect. We compute a fast, non-cryptographic hash of this tensor, $h(\phi) = \text{Hash}(I')$, and maintain a set of all previously seen hash values, $\mathcal{H}$. Before evaluating a new recipe $\phi$, we first check if its hash $h(\phi)$ is present in $\mathcal{H}$. If it is, we prune this candidate and skip its evaluation entirely.

**2. Early Exit during Mask Generation.**  This optimization aims to accelerate the evaluation of a single, unpromising permutation candidate. We maintain a global variable, $|\mathcal{M}_{min}|$, which stores the size of the smallest attention mask (i.e., the best concentration ratio) found so far across all previously evaluated permutations.

During the iterative mask generation process for a new permutation $\phi$ (as described in Section 3.2.2), we continuously monitor the size of the current mask, $|\mathcal{M}|$. If at any step the number of blocks in the mask becomes greater than or equal to the current best, i.e., $|\mathcal{M}| \geq |\mathcal{M}_{min}|$, we can immediately terminate the evaluation for $\phi$. This is because it is impossible for this candidate to yield a better concentration ratio than the one already found. This strategy effectively avoids the cost of completing the iterative expansion for permutations that are clearly suboptimal.

**3. Heuristic Search Order.**  The effectiveness of the early exit strategy is highly dependent on finding a strong initial value for $|\mathcal{M}_{min}|$ as quickly as possible. We empirically observe that the choice of the permute order ($new_shape$) generally has a more substantial impact on the attention map's concentration than the specific tile size.

Based on this insight, we carefully structure the search to prioritize the enumeration of different permutation orders. Our main search loop iterates through all permute order candidates first, while keeping the tile sizes fixed. This heuristic approach significantly increases the probability of discovering a permutation with a small, highly concentrated attention mask early in the search. By establishing a low value for $|\mathcal{M}_{min}|$ at the beginning, this strategy maximizes the pruning power of the early exit technique for the vast majority of remaining candidates in the search space.

## A.2  VISUALIZATION OF THE GENERATED VIDEOS

We provide visual comparisons between Dense Attention and TornadoAttention in Figure 6-13.

Dense Attention

Tornado

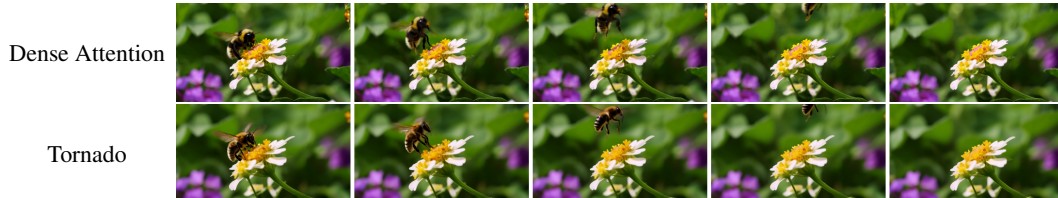

Figure 6: Visualization for dense attention and our method with prompt *In the garden, a little bee was fluttering and then landed on a flower.*

Dense Attention

Tornado

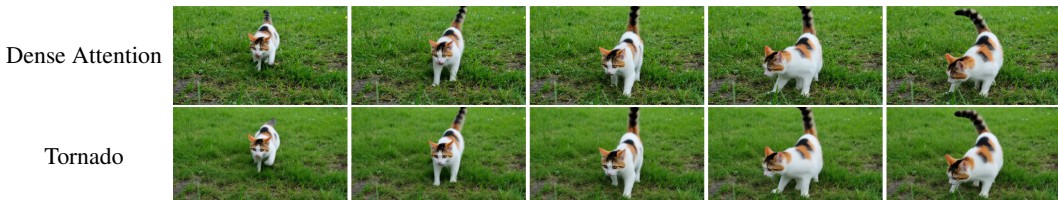

Figure 7: Visualization for dense attention and our method with prompt *A cat walks on the grass, realistic style.*

Dense Attention

Tornado

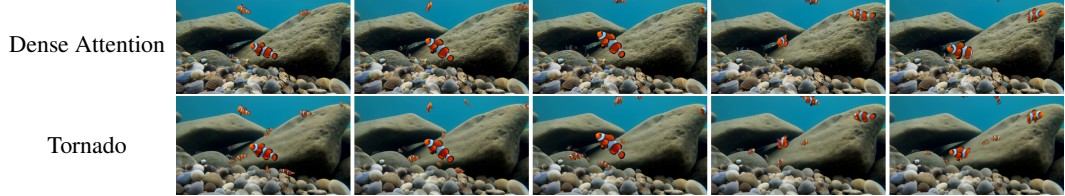

Figure 8: Visualization for dense attention and our method with prompt *Under the rocks in the water, there are many small shells hidden, with numerous clownfish swimming nearby.*

Dense Attention

Tornado

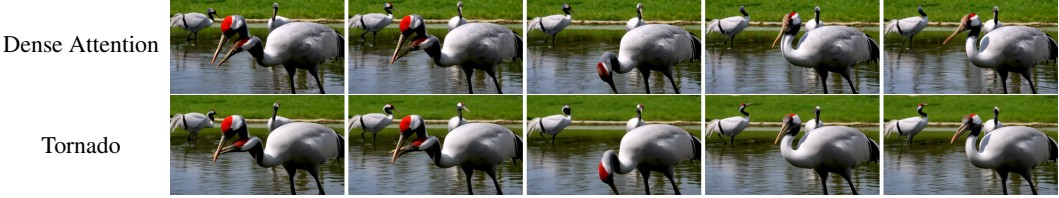

Figure 9: Visualization for dense attention and our method with prompt *By the pond, red-crowned cranes are foraging.Medium shot.*

Dense Attention

Tornado

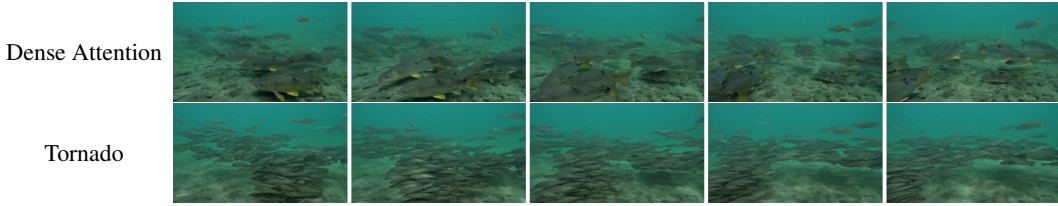

Figure 10: Visualization for dense attention and our method with prompt *A clear lake bottom, where a school of fish leisurely swims.*

Dense Attention

Tornado

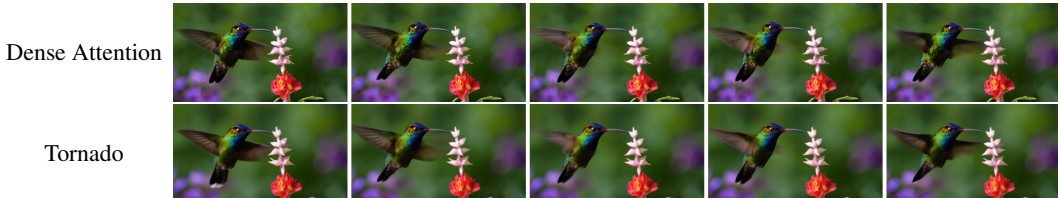

Figure 11: Visualization for dense attention and our method with prompt *A hummingbird flaps its wings and hovers in front of a flower.*

Dense Attention

Tornado

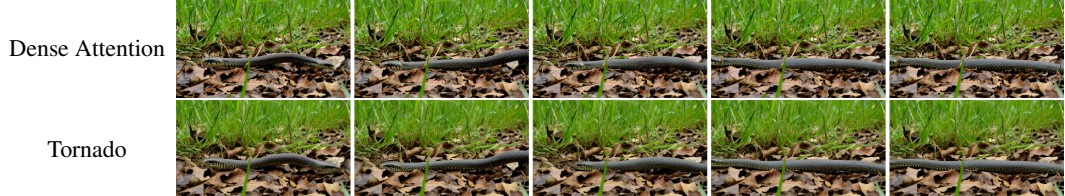

Figure 12: Visualization for dense attention and our method with prompt *A snake slithers smoothly along the ground, weaving agilely through the grass and fallen leaves, tracking shot.*

Dense Attention

Tornado

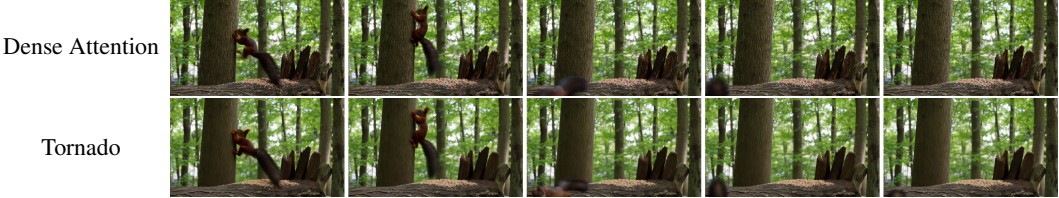

Figure 13: Visualization for dense attention and our method with prompt *A squirrel is busily jumping on a tree trunk, looking for food to prepare for winter.*

