# OpenReview forum: "TornadoAttention: Hardware-Efficient Sparse Attention via Fine-Grained Spatio-Temporal Permutation for Video Generation"
_ICLR.cc/2026/Conference — Submitted to ICLR 2026_

### Official Review · Reviewer_E4qR · 2025-10-29

**Soundness:** 2
**Presentation:** 1
**Contribution:** 1
**Rating:** 2
**Confidence:** 5

**Summary:**

The paper proposes TornadoAttention, a training-free sparse attention mechanism for accelerating video diffusion.

The core idea is to apply a fine-grained spatial temporal permutation over query and key, and reorder tokens to align the memory layout with the natural locality of video data.

The permutation and mask are found via an offline search.

Experimental results show up to 1.4× speedup.

**Strengths:**

1. This paper explore the spatial and temporal redundancy during video generation.

**Weaknesses:**

1. This paper only show the results in Table 2, only one result actually, and the result is worse than SVG.

2. The spatio-temporal permutation is not novel, similar as SVG [1].

3. The  memory layout alignment is similar as DraftAttention [2].

4. Overall paper is bad.


---
[1] Sparse VideoGen: Accelerating Video Diffusion Transformers with Spatial-Temporal Sparsity

[2] DraftAttention: Fast Video Diffusion via Low-Resolution Attention Guidance

**Questions:**

1. The results is bad.

2. The idea is from other works.

---

### Official Review · Reviewer_Qjwj · 2025-10-29

**Soundness:** 2
**Presentation:** 2
**Contribution:** 2
**Rating:** 4
**Confidence:** 4

**Summary:**

The paper proposes TornadoAttention, a training-free sparse attention mechanism for Diffusion Transformers (DiTs) in video generation. The core contribution is a fine-grained spatio-temporal permutation search that reorders query/key tokens so that high-magnitude attention values become concentrated near the main diagonal, enabling block-sparse masking aligned with GPU-friendly kernels. An offline search explores permutations over temporal, height, and width tilings and selects the configuration achieving the highest attention “energy” concentration (top‑p threshold τ). Integrated into the Sparse VideoGen (SVG) stack, TornadoAttention reportedly delivers a 1.4× speedup on HunyuanVideo-T2V-14B with small perceptual degradation, while maintaining competitive VBench metrics.

**Strengths:**

* Originality: Introducing a permutation-based search to align logical locality with memory layout is an interesting twist relative to prior fixed-mask or dynamic-search sparse attention schemes. The idea of using einops-style permutations and block tilings to coax sparse structure is creative.
* Quality: The offline search is described in detail, including pruning, early stopping, and heuristics, providing some reproducibility. Integrating the method into a public codebase (SVG + Magi Attention) demonstrates engineering effort.
* Clarity: The paper clearly motivates the misalignment between rasterized token order and actual spatio-temporal locality, supported by helpful figures and step-by-step descriptions of the search algorithm.
* Significance: If broadly applicable, a training-free, prompt-agnostic sparse mask with substantial speedup would be of practical value for large video DiTs, especially since many production deployments cannot afford fine-tuning.

**Weaknesses:**

* Empirical validation is narrow and partially contradictory. The main quantitative result (Table 2) shows a drop from 31.12 dB to 25.18 dB PSNR and from 0.9337 to 0.8361 SSIM relative to the dense baseline, which is hardly “negligible.” LPIPS also worsens substantially. Yet qualitative claims emphasize negligible quality loss, creating tension. More thorough analysis (e.g., attention reconstructions, human evaluation) is missing. Also missing comparison with similar works like Radial Attention.
* Speedup reporting lacks rigor. The 1.4× figure is based on averaged latency over 10 videos on 8×H100 GPUs, but the paper does not specify variance, batch size, or how much time is spent in kernels versus search overhead. There is no comparison to other recent training-free baselines (e.g., STA, Radial Attention, CompactAttention) under the same hardware. FLOP reductions or roofline analysis are absent.
* Limited baseline coverage. SVG is the only competitor, yet TornadoAttention is implemented within the SVG stack; it is unclear whether SVG’s own static masks are disabled for fair comparison. Without side-by-side evaluation against more recent adaptive sparse methods (DraftAttention, AdaSpa, Jenga) or permutation-based reshaping (PAROAttention), it is hard to judge relative merit.
* Offline search cost and generalization are underspecified. The search reportedly takes up to 30 minutes per layer even after pruning, but the total time over all layers/heads is not given, nor is the dataset or prompt set used to collect proxy attention matrices. Claims of prompt agnosticism need evidence: how sensitive are masks to changes in video duration or resolution?
* Discussion of failure cases is missing. No analysis is provided for scenarios where permutation search fails (e.g., global-head attention), nor any fallback mechanism.

**Questions:**

* Quality degradation versus speedup: How do the authors reconcile the substantial PSNR/SSIM drop with the claim of negligible degradation? Are there scenarios/prompts where degradation is unacceptable? Could a higher τ or mixed dense/sparse scheduling alleviate this, and at what cost to speed?
* Offline search dataset: What corpus of prompts/videos is used to collect attention matrices for the offline search? How many samples per head/layer, and how do you ensure coverage of diverse motion/content?
* Search scalability: What is the total wall-clock time and compute cost to run the search for HunyuanVideo-T2V-14B (all layers/heads)? How does this scale with sequence length or resolution?
* Generalization failsafes: If the model encounters a scene whose attention patterns differ significantly from the search set, does performance degrade? Have you tested on different resolutions (e.g., 1080p) or on different DiTs beyond the listed ones?
* Comparison to alternative permutations: Can you provide quantitative comparisons (quality and latency) against relative methods?

---

### Official Review · Reviewer_XQYT · 2025-10-30

**Soundness:** 3
**Presentation:** 3
**Contribution:** 3
**Rating:** 4
**Confidence:** 3

**Summary:**

This paper proposes TornadoAttention, a training-free sparse attention mechanism for accelerating video Diffusion Transformers (DiTs). The key innovation is applying fine-grained spatio-temporal permutations to reorder token sequences, concentrating attention scores around the main diagonal. Through offline search over permutation orders and tile granularities, the method identifies optimal static sparse masks for each attention head. Evaluated on HunyuanVideo and Wan2.1 models, TornadoAttention achieves 1.4× speedup with minimal quality degradation.

**Strengths:**

The permutation-based approach is novel - reordering tokens to align memory layout with spatio-temporal locality differs from existing fixed-pattern or dynamic sparse attention methods. The 4D search space (permutation order plus T/H/W tile sizes) provides a principled framework for exploring video data structure. The paper clearly motivates the problem with examples of diverse attention patterns that defy simple classification. The visualization of misalignment between logical locality and memory layout effectively illustrates why standard token ordering is suboptimal. Evaluation on large-scale state-of-the-art models (HunyuanVideo-14B, Wan2.1-14B) demonstrates practical applicability. The method maintains good fidelity metrics (PSNR, SSIM, VBench) while achieving measurable speedup, and the ablation validates the necessity of the full search space.

**Weaknesses:**

1.The paper only compares against one baseline (SVG). Missing comparisons with other recent sparse attention methods such as MInference, DiTFastAttn, PAB, DraftAttention, Radial Attention, CompactAttention, and AdaSpa that are discussed in related work.

2.Ablation studies are insufficient. Only one ablation on the 4D search space is provided. Critical design choices lack analysis: the energy threshold τ=0.9, block size b=128, number of profiling samples, and the decision to skip first 25% denoising steps.

3.Model coverage is narrow. Evaluation limited to two models (HunyuanVideo and Wan2.1). Missing validation on other video DiT architectures like CogVideoX which is widely adopted in the community.

4.Quality metrics are incomplete. Reports only PSNR, SSIM, LPIPS and two VBench metrics. Missing comprehensive video quality evaluation including temporal consistency, motion smoothness, aesthetic quality, and other VBench dimensions that are standard for video generation assessment.

5.No end-to-end latency breakdown is provided. The paper reports overall 1.4× speedup but lacks detailed analysis of where improvements come from - attention computation, memory access, kernel overhead, or other components. Missing kernel-level performance analysis showing actual performance of the permutation strategy.

**Questions:**

See weakness.

---

### Official Review · Reviewer_YMQr · 2025-11-01

**Soundness:** 3
**Presentation:** 2
**Contribution:** 2
**Rating:** 4
**Confidence:** 4

**Summary:**

This paper introduces TornadoAttention, a training-free sparse attention mechanism for Diffusion Transformers (DiTs) in video generation. The core idea is to reorder the spatiotemporal token sequence via a fine-grained permutation search, allowing attention mass to concentrate near the diagonal. Experiments on SOTA models like Hunyuan Vedio-T2V-14B show a 1.4x reduction in computational load.

**Strengths:**

* The motivation behind the proposed method is well articulated.

* Being training-free and prompt-agnostic, the proposed approach has low integration cost.

**Weaknesses:**

* No formal justification that the proposed permutation space is sufficient or near-optimal.

* The evaluation is conducted on a few large-scale DiT models (Hunyuan-T2V-14B and Wan2.1-14B). While these are SOTA, they represent a very similar architectural class.

* The method is only compared against SVG, missing comparisons to more recent and stronger sparse attention techniques.

**Questions:**

1.The evaluation on Wan2.1 is claimed but never shown. Please clarify whether these experiments were actually conducted.

2.Was the TornadoAttention method tested on other, more diverse DiT architectures, such as smaller-scale models or those with different configurations？

3.The resulting 1.4x speedup is a marginal improvement, and its effectiveness is not convincingly demonstrated, as it is only compared against a single baseline (SVG).

4.All experiments are conducted at 720p. Have the authors evaluated longer videos or higher resolutions, where spatio-temporal sparsity patterns may differ significantly?

---

### Meta-Review · Area_Chair_2s5q · 2025-12-27

**Summary:**

This paper proposes a training-free sparse attention for accelerating video diffusion transformers by permuting tokens to concentrate attention scores around the diagonal for block-sparse masking. However, all reviewers raised concerns about the insufficient and partially contradictory empirical validation, where only one result on one model against one baseline is available, and shows non-negligible quality drops. No rebuttal was submitted, and no discussion occurred.

**Reviewer Concerns:**

N/A, as no rebuttal was submitted

**Reviewer Scores:**

N/A, as no rebuttal was submitted

---

### Decision · Program_Chairs · 2026-01-26

Reject